# Stress and Deformation Analysis of Prestressed Wound Composite Components with an Arch-Shaped Metal Liner

**DOI:** 10.3390/ma17030757

**Published:** 2024-02-05

**Authors:** Junsheng Wang, Jun Xiao, Dajun Huan, Lei Yan, Zijie Wang, Zhiwei Tao

**Affiliations:** R&D Center for Composites Industry Automation, College of Material Science and Technology, Nanjing University of Aeronautics and Astronautics, Nanjing 210016, China; js_wong@nuaa.edu.cn (J.W.); huandj@nuaa.edu.cn (D.H.); bx2006315@nuaa.edu.cn (L.Y.); zj-wang06083@nuaa.edu.cn (Z.W.); zwtao233@163.com (Z.T.)

**Keywords:** ultra-high prestressed winding, arch-shaped section, additional bending moment effect, finite element analysis

## Abstract

The stress distribution in prestressed filament wound components plays a crucial role in determining the quality of these components during their operational lifespan. This article proposes a physical model to analyze the stress and deformation of prestressed wound composite components with arch-shaped sections. Drawing upon the principles of beam theory, we delve into the analysis of prestressed wound components with metal liners featuring arch-shaped sections. Our investigation revealed a noteworthy phenomenon termed the “additional bending moment effect” within prestressed wound components with arch-shaped sections. Furthermore, this study establishes a relationship between this additional bending moment and the external pressure. In addition, a 3D finite element (FE) model for prestressed wound components with arch-shaped sections incorporating metal liners was developed. The model’s accuracy was validated through a comparison with prestressed wound experiments, showcasing an error margin of less than 2%. In comparison with prestressed wound components with circular cross-sections under identical load and dimensional parameters, it was observed that prestressed wound components with arch-shaped sections exhibit stress distributions in the arc segments akin to their circular counterparts, with differences not exceeding 5%. Notably, when the ratio of the straight segment length to the inner diameter of the arc segment inner is less than 4, the deformation on the symmetric plane of the arc segment in an arch-shaped component can be effectively considered as the summation of deformations in equivalent-sized arc and straight segments under identical loading conditions. This yields an equivalent physical model and a streamlined analysis and design methodology for describing the deformation characteristics of prestressed wound components with arch-shaped sections.

## 1. Introduction

Owing to their exceptionally high specific strength, modulus, excellent corrosion resistance, and versatile design attributes, composite materials have witnessed widespread applications in diverse sectors such as energy, military, and aerospace in recent years. Filament winding technology stands out as a widely employed, cost-effective manufacturing method in shaping composite materials. Notably, recent advancements in materials and equipment have enabled an increase in the winding tension in filament winding technology, reaching up to 50% of the material strength [1]. This innovation has given rise to the emergence of prestressed filament winding. This technology finds particular relevance in the production of specialized products, including surface-mounted motor rotor sheaths [2], energy storage flywheels [3,4,5,6], and electromagnetic railgun barrels [7], among others. For these products, the stress–strain state following winding significantly impacts their operational performance, necessitating a thorough analysis of stress and strain in highly prestressed winding components. Furthermore, it is important to highlight that, in contrast to traditional filament wound components, prestressed wound components often manifest as thick-walled structures. Predictive models based on conventional thin-shell theory are inadequate [8,9]. Hence, there is an imperative need for analytical approaches specifically tailored to the characteristics of thick-shelled structures.

At present, many scholars have undertaken analyses of the post-winding residual stresses in filament-wound thick-shelled components. Some researchers employ statistical methods to investigate how process parameters of winding affect the stress distribution in components, emphasizing the critical role of winding tension as a significant parameter affecting component stress [10,11,12,13]. Others concentrate on utilizing analytical models to scrutinize and forecast the distribution of stresses during the winding process. For instance, Lee and Springer [14,15] equivalently represented wound components as several elastically superimposed thin circular rings, establishing a stress–strain model for wound components. Lu et al. [16] established a theoretical model to study the residual stresses of wound components developed during the filament winding process. The model contains four sub-models: fiber motion, thermal, rheological, and stress–strain. The model was validated by experimentation and both the results from the theoretical model and experimentation indicated that winding tension has a large influence on residual stresses, with the authors concluding that it is a suitable way to control residual stresses by adjusting the winding tension. To investigate the residual stresses of in situ filament winding of a thermoplastic matrix composite, Dedieu et al. [17] proposed an analytical model to predict process-induced residual stresses. The model considers the influence of temperature, continuous bonding during the process, the processed layer on the structures, and the curvature of the mandrel, and its pertinence was verified by a comparison between measured and computed end-to-end openings of split rings. These models from the references consider the influence of several process parameters on the residual stresses. However, for prestressed winding, tension is the key variable on residual stresses. Kang et al. [18], based on anisotropic elasticity theory and isotropic thick-walled cylinder theory, derived the radial and circumferential stresses in a winding layer under external pressure in relation to winding tension. They established an analytical model describing the relationship between winding tension and component stress distribution based on the superposition principle. Liu and Shi [19], employing three-dimensional thick-walled cylinder elasticity theory and superposition principles, established an analytical model for stress induced by winding tension in components. They considered the relaxation effects of the mandrel and previous wound layers, developing an iterative algorithm for computing residual stresses based on the reverse iteration principle. Geng and Wang [20], using superposition and reverse unloading principles, established analytical expressions for the distribution of winding tension and component stress, demonstrating agreement with their experimental and finite element results. Nevertheless, the models derived from the aforementioned literature are all based on the principle of elastic superposition. Our research team discovered that when the ratio of the thickness of the single layer to the outer diameter of the winding liner is large, analytical models based on the principle of elastic superposition can result in substantial errors [21]. To mitigate this error, our research team, based on thick-walled cylinder theory and anisotropic elasticity theory and utilizing an inverse iteration algorithm, established a prestress algorithm for circular section wound components containing metal liners. We analyzed the factors influencing the differences between this algorithm and those based on the elastic superposition principle.

In addition, the finite element method is another common way to analyze the stress and deformation distribution of wound components. Most of the finite element models were established to simulate the actual filament winding process [22,23]. However, for prestressed winding, the way to simulate tension application and layer-by-layer steps are more significant. Ren et al. [24] employed a comprehensive finite element analysis method that combines real and fictitious elements along with a layer-by-layer solidification process to simulate the winding procedure. They established an iterative algorithm for the design of residual stresses in the winding layer of an internal pressurized vessel. Zhang et al. [25] introduced a finite element method that leverages equivalent cooling to simulate prestress application. This method treats the prestress induced by winding tension as equivalent to the prestress resulting from the cooling of composite layers. The relationship between the cooling amount and prestress was provided, facilitating an exploration of the impact of winding tension on wound gas cylinders. Cheng [26], employing the equivalent cooling method in conjunction with birth-and-death element technology and element tracking techniques, successfully conducted a numerical simulation of the stress distribution in thermoplastic composite wound components with prestress.

Nevertheless, the studies cited above primarily focused on components with circular cross-sections. Limited research has been reported for non-circular cross-section components, especially those featuring arch-shaped sections. Root et al. [27] conducted a comparative study using the finite element method to analyze the rail expansion of different railgun barrels with non-circular cross-sections under the influence of winding prestress. They also assessed the merits and drawbacks of various cross-sectional shapes. Zu [28], employing the finite element method, investigated the stress distribution in prestressed filament wound components with arch-shaped sections, confirming the effectiveness of the employed model. Our research team further analyzed the influencing factors on the deformation of prestressed composite barrels with arch-shaped sections during the loading of an electromagnetic railgun [29].

Based on the analysis above, it is evident that the current research on the stress distribution of prestressed wound components is relatively comprehensive. However, this research has primarily focused on components with circular cross-sections, leaving the analysis of components with arch-shaped sections insufficiently explored. Notably, the findings from Refs. [27,29] indicate that when the cross-sectional shape of an electromagnetic railgun barrel is arch-shaped, the rate of rail expansion is lower, leading to improved launch efficiency and accuracy. As a result, there is a critical need for a more thorough examination of the stress distribution in prestressed wound components with arch-shaped sections. This analysis is crucial for advancing the practical application of prestressed winding processes in engineering.

To investigate the stress distribution in prestressed wound components with arch-shaped sections, this study, based on a plane beam model, analyzes the stress distribution when only the convex surface was subjected to pressure. Through finite element simulation and experimental methods, the stress distribution and deformation of the component during the winding process were examined. By comparing prestressed wound components with circular cross-sections, the inherent correlation between the stress and deformation of the two was explored, leading to the development of a physical model for predicting the stress and deformation of prestressed wound components with arch-shaped sections.

## 2. Materials and Methods

The arch shape consists of two semi-circular rings and two straight segments in the middle, as illustrated by Figure 1. In this figure, P(θ) represents the external pressure on the liner generated by the winding tension in the wound layer; $\sigma_{ir}(\theta )$ and $\sigma_{i\theta }(\theta )$ denote the radial and circumferential stresses on the component before the winding of the new layer; $T_{j}$ represents the winding tension in the new layer; while $P{\left(\theta \right)}^{\prime }$, $\sigma_{ir}{\left(\theta \right)}^{\prime }$, and $\sigma_{i\theta }{\left(\theta \right)}^{\prime }$ represent the external pressure on the liner, as well as the radial and circumferential stresses on the component after the winding of the new layer. According to the principles of winding, winding tension transforms into pressure only on the convex surface of the liner. For the liner of the arch-shaped section, only the outer surface of the arc segment generates external pressure due to winding tension (Figure 1), and this pressure is associated with the central angle. To analyze the stress distribution and deformation of prestressed wound arch-shaped section component, three methods were employed, including analytical, numerical, and experimental methods.

### 2.1. Stress Distribution of the Prestressed Wound Metal Liner

As mentioned above, for the prestressed wound liner, only the convex surface suffers the pressure generated by prestress, as shown in Figure 1a. To analyze the stress distribution of the prestressed wound liner, the plane beam model was used, and we made the following assumptions:Plane assumption—The transverse cross-section is initially a plane before deformation, remains a plane after deformation, and continues to be perpendicular to the deformed axis;End effects are neglected, and the component is assumed to be infinitely long;Axial stress is neglected, and stress along the axial direction is assumed to be uniformly distributed;The applied pressure is a function of the central angle θ, i.e., p=f(θ).

For convenience, a quarter of the liner was taken for analysis, as shown in Figure 2.

In Figure 2, p=f(θ) represents the external pressure. For any section, m−m corresponds to the straight segment AB, and according to static equilibrium, several equations can be obtained as the following:(1)$$\begin{array}{l} F_{A}^{N}=F_{X}^{N}\\ F_{A}^{S}=F_{X}^{S}\\ M_{A}=M_{X} \end{array}$$

Similarly, for any section, n−n corresponds to the arc segment BC, and the following equations can be obtained.
(2)$$\begin{array}{l} F_{A}^{N}+F_{\varphi }^{S}\sin\varphi =F_{\varphi }^{N}\cos\varphi +\int_{0}^{\varphi }pR_{2}\sin\varphi d\varphi \\ F_{A}^{S}+F_{\varphi }^{S}\cos\varphi +F_{\varphi }^{N}\sin\varphi =\int_{0}^{\varphi }pR_{2}\cos\varphi d\varphi \\ M_{A}+\int_{0}^{\varphi }pR_{2}R_{0}\sin(\varphi -\theta )d\theta =M_{\theta }+F_{A}^{N}R_{0}(1-\cos\varphi ) \end{array}$$

In the equations, the superscripts indicate the type of force, with *N* and *S* representing axial force and shear force, respectively. The subscript *X* denotes the position along the straight segment *AB* at a distance of *X* from point A, and φ represents any central angle of the arc segment. $R_{0}$ represents the neutral axis radius, since the section is rectangular, $R_{0}=(R_{1}+R_{2})/2$, and $R_{1},R_{2}$ are the inner and outer radii of the arc segments from the arch-shaped section. By substituting this into Equations (1) and (2), the following relationship can be derived.
(3)$$\begin{array}{l} F_{A}^{S}=F_{X}^{S}=F_{\varphi }^{S}\\ F_{A}^{N}=F_{X}^{N}=F_{\varphi }^{N}\\ M_{A}=M_{X}=M_{\varphi } \end{array}$$

As shown in Figure 2, the central angle is θ=π/2, so the following equation can be derived.
(4)$$F_{A}^{N}=\int_{0}^{\pi /2}pR_{2}\sin\varphi d\varphi$$

Based on Equations (1)–(4), it is evident that if the pressure resulting from the winding tension is known, all other mechanical parameters can be determined, and the bending moment $M_{A}$ can be calculated using Castigliano’s theorem, which can be expressed as
(5)$$V_{\varepsilon }=\int_{S}(\frac{M^{2}}{2ESR_{0}}+\frac{MF_{N}^{2}}{EAR_{0}}+\frac{F_{N}^{2}}{2EA}+\frac{kF_{S}^{2}}{2GA})ds$$
where $V_{\varepsilon }$ represents the strain energy; $M,\;F_{N},\;F_{S}$ denote the bending moment, axial force, and shear force, respectively; E, G represent the elastic modulus and shear modulus; $S,\;A,\;R_{0}$ represent the static moment, the sectional area, and the neutral axis radius, respectively; and $S=A(R_{0}-r_{s})$ with $r_{s}=\frac{b}{\ln\frac{R_{1}}{R_{2}}}$ (b represents the wall thickness of the arch-shaped section). k is a factor associated with the section shape.

Due to symmetry, it is known that section A remains unrotated after deformation, meaning the angle $\phi_{A}$ is 0. Thus, in section A, applying Castigliano’s theorem, we obtain the following equation:(6)$$\phi_{A}=\frac{\partial V_{\varepsilon }}{\partial M_{A}}=0$$

By combining Equations (1)–(6), the bending moment $M_{A}$ can be determined, and it is a function related to p. Consequently, the circumferential stress of the arch-shaped section under external pressure solely on the arc segment can be derived.
(7)$$\sigma_{\theta }=\frac{My}{Sr_{s}}+\frac{F_{N}}{A}$$

In summary, the analytical model above establishes the relationship between circumferential stress and external pressure analytically, but lacks a solution for radial stress. Furthermore, the model highlights that the surface pressure, generated by tension, is related to the central angle θ, serving as an unknown variable. Thus, the stress distribution cannot be solved analytically. Subsequently, the following sections will use finite element methods to scrutinize the stress and deformation of prestressed wound components with a metal liner featuring an arch-shaped section. Despite the limitations of the analytical model in calculating stress distribution, the analysis above reveals the emergence of an additional bending moment $M_{A}$ due to the non-rotational symmetry of the structure. This bending moment induces deformations in the straight segment of the arch-shaped section, causing the arc segment to become “non-circular”. The curvature radius of the static neutral layer emerges as a crucial factor influencing the stress distribution in the structure.

### 2.2. FE Modelling

In Section 2.1, we established a plane beam model to analyze the stress distribution of the prestressed wound arch-shaped metal liner, finding out that the stress distribution cannot be solved analytically. Therefore, we established a 3D FE model to simulate the winding process of the prestressed wound component, as shown in Figure 3. For comparison with the simulation results of the prestressed wound component with a circular section in the literature [21], the radii of the arc segment in the arch-shaped section, as well as the inner and outer radii of the liner for the circular section, remained the same. This facilitates comparisons with subsequent experiments. The winding angle is 90°. It is worth noting that an end effect plays a significant role when the axial length of the arch-shaped component is relatively small [30]. Thus, the dimensions of the component are shown in Figure 4, and its parameters related are outlined in Table 1. How this end effect influences the deformation of the component will be discussed in Section 3.

The elements used throughout the model are 3D incompatible elements (C3D8I). The length of the liner element is approximately 2 mm, and the winding layer elements share the same length as the liner elements, excluding their thickness. The length in the thickness direction aligns with a single winding layer, which is 0.14 mm. To facilitate comparison with subsequent experiments, the FE model incorporates the core shaft and the baffles essential for the winding experiment. Connections between the baffles and the liner, between the baffles and the core shaft, and between the winding layer and the liner are all tied constraints. Pinned constraints are applied to the end face of the core shaft. The material of the liner in the model is Q235, and the material used for the winding layer is the carbon fiber-reinforced thermoplastic composite AS4D/PEEK. The properties of the two materials are listed in Table 2 and Table 3. The winding layer’s ply direction is defined as the following: Direction 1 is axial, Direction 2 is circumferential along the outer contour of the arch-shaped component, and Direction 3 is the stacking direction of the winding layers (Figure 3). The materials of the baffles and the core shaft are also Q235. Additionally, during the finite element analysis, the simulation of the prestress application and layer-by-layer stacking procedure incorporates the equivalent cooling method, birth-and-death element method, and real and fictitious element strategy mentioned by Ref. [21].

### 2.3. Experimental Setup

Referring to the prestressed winding experiment we previously set up in the literature [21], we conducted a validation experiment for prestressed winding of arch-shaped components with a metal liner to assess the accuracy of the FE model and discuss the stress and deformation of the prestressed wound arch-shaped component with metal liner.

Similar to the prestressed winding experiment for circular components with a metal liner, this experiment used strain gauges to monitor strain values at specific points on the inner wall of the liner. A comparison was then made with the strain values obtained from the finite element simulations to validate the accuracy of the model. The procedure involved attaching strain gauges to the inner wall of the arch-shaped metal liner (Figure 4). Subsequently, the metal liner was mounted in the winding equipment (Figure 5), following the rule that the central axis of the compact roller always coincided with the normal vector to the outer surface of the liner (Figure 6). A trajectory program was written, and instant adhesive was used between layers to avoid the influence of thermal loads and resin rheology on the stress distribution of the component [21]. Deformation data for the liner were collected using a wireless strain acquisition system. The experiment used the notation A–C to represent the positions of the strain gauges; for example, 1-1 denotes the axial position 1 and circumferential position 1 of the strain gauge. The winding tension was set at 400 N, and the experiment was conducted for a total of 16 layers in 4 iterations.

## 3. Results and Discussion

This section will analyze the experimental results and compare them with the outcomes of the FE model. Additionally, we will discuss the distinctions and correlations between arch-shaped and circular prestressed wound components.

### 3.1. Experimental Results and Validation of the FE Model

The experiment was conducted four times. Due to the symmetry of the arch-shaped component, the strains at symmetric positions were averaged. Specifically, averaging was performed between positions 1-1 and 1-5, positions 1-2, 1-4, 1-6, and 1-8, positions 1-3 and 1-7, positions 2-1 and 2-5, positions 2-2 and 2-4, 2-6, and 2-8, and positions 2-3 and 2-7, as shown in Figure 4. Thus, the results were obtained, as illustrated in Figure 7. The averaged results are denoted by the superscript A.

From Figure 7, it is evident that the differences in the experimental results between axial positions 1 and 2 are less than 2%, proving that the studied positions were not significantly affected by end effects. Among the three circumferential positions monitored, position 3 exhibited a larger strain value, while positions 2 and 1 had comparable strain values. This indicates that the pressure on the liner surface, resulting from the winding tension, was not uniformly distributed circumferentially along the arch-shaped section. Additionally, the maximum deformation occurred at the apex of the arch-shaped section, attributed to the bending deformation of the straight segment under the influence of bending moments, thus leading to the “non-circular” symmetry of the arch-shaped section.

Using the ABAQUS post-processing program, the strain values corresponding to the positions where the strain gauges were attached in the experimental setup were extracted from the finite element simulation results of the prestressed winding for arch-shaped components. These values were then compared with the experimental results, as illustrated in Figure 7.

The curves obtained from the experimental values and finite element simulation values in Figure 7 almost perfectly overlap, with the error in the finite element values being less than 5%. This validates the accuracy of the FE model. Furthermore, within the range of 16 winding layers, the strain values at various measurement points on the inner wall of the arch-shaped section liner exhibit an approximately linear relationship with the number of winding layers. In contrast, as indicated by the literature [21], for circular components, the change in inner wall strain is non-linear with the number of winding layers. This non-linearity is attributed to the weakening effect of the surface pressure converted from winding tension on the liner as the number of winding layers increases, known as the elastic relaxation effect. Figure 7 illustrates that the strain changes at various measurement points on the inner wall of the arch-shaped section liner are quite similar. Therefore, the strain increment at position 1-3 (apex of the arch-shaped section) after completing the winding at each layer was plotted, and a linear fitting was applied to generate Figure 8.

From Figure 8, it is evident that the strain increment (absolute value) at position 1-3^A^ gradually decreases with the increase in the number of winding layers. Conducting a linear fitting resulted in the equation y=0.11235x−21.5925, where the $R^{2}$ for this line is 0.975 and the value of Pearson’s r is 0.988. These high correlation coefficients indicate that the obtained line fits the corresponding data well. Therefore, it is clear that the influence of winding tension on the liner decreases with the increasing number of winding layers during the prestressed winding process of the arch-shaped section, indicating the presence of an elastic relaxation effect in the prestressed winding process of the arch-shaped component.

### 3.2. Stress Analysis of the Prestressed Wound Arch-Shaped Component with a Metal Liner

The experimental and finite element methods discussed earlier for the prestressed wound arch-shaped component with a metal liner are time-consuming. However, the results from 3D finite element simulations and experiments show that when the axial length is sufficiently long, the end effects of the arch-shaped component can be neglected. Therefore, it becomes feasible to simplify the 3D problem into a 2D one for analysis. In this section, a 2D FE model for the prestressed wound arch-shaped component will be developed and compared with the simulation results of the 3D model. Additionally, we will explore the differences and connections between the stress and deformation of the 2D arch-shaped section and circular prestressed wound components.

#### 3.2.1. Establishing a 2D FE Model for the Prestressed Wound Arch-Shaped Component

To facilitate comparison with the 3D model, we took a section of the 3D model for the construction of a 2D model. The dimensional and material parameters were kept consistent with those of the section in the 3D model. The element type used is the plane stress-reduced integration element CPS4R, with a liner element size of 1 mm^2^. The elements are refined on both the inner and outer surfaces (Figure 9). The element length of the winding layer is 1 mm, and its thickness corresponds to the actual single-layer thickness. Combined with the equivalent cooling method, birth-and-death element method, and real and fictitious element strategy, the model simulates the prestressed winding process of the arch-shaped component.

#### 3.2.2. Comparison of the 2D and 3D Model Results

The strains at the experimental measurement points in the 2D model after each winding layer were extracted and compared with the results from the 3D model, as shown in Figure 10. Due to their structural symmetry, only positions 1-1, 1-2, and 1-3 were considered for comparison.

From Figure 10, it is evident that the simulation results of the 2D and 3D models are nearly identical. For position 1-2, the maximum error within the studied range is less than 5%, falling within an acceptable range. This result suggests that the influence of the ends on the studied points can be neglected in the 3D model employed. Figure 11 illustrates the simulated deformations of the 3D and 2D models. Observing Figure 11, the 3D model exhibits a “pseudo-hourglass” shape in the axial direction after deformation (Figure 11b), with the straight segment appearing “inward concave”, in contrast to the “outward convex” shape of the straight segment in the 2D model. Additionally, considering Figure 2, Figure 7 and Figure 10, it is clear that the Y-direction strain in the straight segment is negative. The surface pressure formed by the tension in the arc segment manifests as a bending moment in the straight segment, and this bending moment leads to an “outward convex” deformation. Therefore, the contour of the axial midpoint, 40 mm, 80 mm, 97 mm, and 115 mm from the centerline, and the end section of the 3D model were analyzed, as depicted in Figure 12.

From Figure 12, it can be observed that at the centerline and a distance of 40 mm from it, the deformation of the straight segment is “outward convex”, and the deformation magnitude is comparable to the 2D model. At positions 80 mm and 97 mm away from the centerline, the “outward convex” curvature of the straight segment gradually decreases. At the position 115 mm away from the centerline, the deformation changes from “outward convex” to “inward concave” and extends along the axial direction. The curvature of the “inward concave” gradually increases until the end. The reason for this result is that the mold is connected to the core shaft through screws, limiting the deformation of the straight segment at the end. At the same time, there is a 9 mm non-winding region at both ends of the liner (screw-in part). The pressure generated by the winding tension in the wound part forms a force couple in the non-wound part, causing the arc segment of the non-wound part to flip in the XZ plane (Figure 11), thereby causing the straight segment to flip and extend. Due to the fixed screws, this ultimately leads to the end of the straight segment being “inward concave”. This result also indicates that the end effect caused by the connection method is more pronounced in the region close to the end during the winding process of the arch-shaped component.

#### 3.2.3. Analysis of Stress Distribution in the Arch-Shaped Prestressed Wound Components with a Metal Liner

As per the insights from the previous sections, when the axial length of the liner is sufficient, the influence of the end can be neglected for positions far from the end. The 2D model proves effective in simulating the prestressed winding process of the arch-shaped component. As mentioned in Section 2.1, obtaining the stress distribution of arch-shaped components through analytical methods is challenging. In this section, a 2D FE model is employed to analyze the stress distribution of arch-shaped prestressed wound components with a metal liner during the winding process. The results are then compared with the analysis of the circular components.

Figure 13 illustrates the circumferential stress variation at different measurement points during the winding process with a winding tension of 500 N after 16 layers for different straight section lengths ($L_{s}$) of the arch-shaped liner. Observing Figure 13, it becomes evident that the circumferential stress at each measurement point decreases with an increase in the straight segment length. As discussed in Section 2.1, the straight segment primarily experiences axial force derived from the surface pressure of the arc segment. Consequently, the axial force is not uniformly distributed across the straight segment’s thickness, leading to additional bending moments. These additional bending moments decrease with the extension of the straight section length (Table 4), resulting in reduced stress within the straight segment as its length decreases (Figure 13a). Regarding stress in the arc segment (Figure 13b,c), it is influenced by the increased length of the straight segment, leading to enhanced overall liner stiffness and reduced deformation, consequently causing a decrease in stress.

The transformation of winding tension into external pressure on the liner is a critical parameter in the prestressed winding process. For the arch-shaped liner, owing to its non-circular circumferential symmetry, the pressure resulting from the winding tension on its arc segment does not distribute uniformly along the circumferential direction. Figure 14 illustrates the contour of radial stress distribution for the arc segment of the arch-shaped section and the circular section after winding 30 layers under a tension of 500 N. From Figure 14, it is evident that the radial stress distribution along the circumferential direction of the arc segment in the arch-shaped section is relatively uniform, except in the transition zone between the arc segment and the straight segment, where the distribution is uneven. Simultaneously, the maximum radial stress occurs at the apex of the arc segment. According to continuity, the radial stress distribution along the circumferential direction should gradually decrease from the apex of the arc toward both sides, displaying a “higher in the middle, smaller at both ends” pattern. Moreover, based on continuity, the surface pressure generated by winding on the outer surface of the arch-shaped liner’s arc segment aligns with the radial stress distribution, gradually decreasing along the circumferential direction from the middle to both sides, i.e., “non-rotational symmetry”. Comparing this with the stress distribution of the circular section, we found that the radial stress gradually decreases with the increase in the length of the straight segment, although the change is very minor. The circumferential stress follows a similar distribution pattern (Figure 15). Additionally, Figure 13 and Figure 14 indicate that the maximum circumferential stress occurs at position 1-2, i.e., the transition point between the straight segment and the arc segment, signifying that the critical point of the arch-shaped prestressed winding structure is at this transition point. Furthermore, from Figure 13c, it was observed that the circumferential stress on the inner surface at the apex of the arc of the arch-shaped liner is not significantly different from the circumferential stress on the inner surface of the circular section. The error is less than 5%, and it decreases with the increase in winding layers.

Based on the preceding analysis, it is evident that the primary forces acting on the straight segment are bending moments and axial forces. The stress analysis revealed that the stress along the *X*-axis direction in the straight segment was nearly 0, as shown in Figure 16. Figure 16 illustrates that *X*-axis stress is only present near the transition from the arc segment to the straight segment, with higher stress on the outer surface compared to the inner surface. Considering stress continuity, it can be inferred that, along the transition line, the *X*-axis stress in the straight segment is continuous with the radial stress in the arc segment. Similarly, on the transition line, the *Y*-axis stress in the straight segment is continuous with the circumferential stress in the arc segment (Figure 17). Figure 17 demonstrates that the straight segment is generally under compression, with the maximum stress in the *Y*-axis direction decreasing slightly as the length of the straight segment increases. Even with an increase in straight segment length from 30 mm to 240 mm, the maximum stress only decreases from 156.9 MPa to 152.0 MPa, which is a change that can be considered negligible.

The stress distribution in the winding layer is also a crucial aspect to consider. Similarly to the circular section, the stress at the inner wall of the first winding layer was chosen for study, and its variation at the same positions as the measurement points in Section 2.3 is shown in Figure 18. From Figure 18, it can be observed that, compared to the circular section, both the radial and circumferential stresses in the winding layer of the arch-shaped component decreased. In Figure 18a, the radial stress at position 1-1 is 0 because position 1-1 is the midpoint of the straight segment, and the winding tension does not convert into pressure in the straight segment. The radial stress at position 1-3 is close to the radial stress obtained for the circular section. The radial stress at position 1-2 shows non-linearity earlier, as position 1-2 is located in the transition zone between the straight segment and the arc segment, which is a geometrically abrupt area, resulting in non-linearity. For circumferential stress, as shown in Figure 18b, the circumferential stress at the three positions is very similar and does not differ significantly from the circumferential stress obtained for the circular section.

Figure 19, Figure 20 and Figure 21 depict the impact of the straight segment length on the stresses in the winding layer. It can be observed from the figures that both the circumferential stress and radial stress decrease with an increase in the length of the straight segment. However, this influence is minor and can be neglected in engineering applications. It is worth noting that, as indicated in Figure 18a, the winding layer on the straight segment (position 1-1) has zero stress in the thickness direction (“radial”). However, its circumferential stress is the same as that at positions 1-2 and 1-3, and with an increase in the number of winding layers, its circumferential stress will decrease. As analyzed by Ref. [21], the reduction in the circumferential stress in the winding layer during the prestressed winding process is due to the elastic relaxation effect caused by the deformation of the already wound layers and the liner under the tension of the new winding layer. The earlier analysis pointed out that there is no pressure on the outer surface of the straight segment, and since the composite material in a single winding layer is continuous, it follows the principle of stress continuity. Therefore, the winding layer on the straight segment also exhibits elastic relaxation effects, although this effect is milder than at positions 1-2 and 1-3.

### 3.3. Deformation Analysis of the Arch-Shaped and Prestressed Wound Component with a Metal Liner

For the circular components, their deformations are of a circular symmetry. As mentioned in the previous analysis, due to the non-circular symmetry of the structure, the stress distribution in the arch-shaped component is not uniform when only the arc segment is subjected to external pressure. Moreover, it is influenced by the length of the straight segment. From this, it can be inferred that its deformation is not rotationally symmetric. Figure 22 illustrates the radial displacement contour of the arc segment of the arch-shaped section with different lengths of straight segments.

From Figure 22, it is evident that, compared to the radial and circumferential stresses, the “non-rotational symmetry” of radial displacement becomes more pronounced with an increase in the length of the straight segment. Due to structural symmetry, the displacement at the midline of the straight segment is subject to the constraint ${u_{x}}^{S}(y=0)=0$. According to continuity, in the transition zone between the straight segment and the arc segment, the displacement is subject to the condition ${u_{x}}^{S}(y=L_{S}/2)={u_{r}}^{R}(\theta =0)$, where the superscript *S* represents the physical quantity of the straight segment, *R* represents that of the arc segment, $L_{S}$ is the length of the straight segment, and the subscript corresponds to the respective coordinate axis. As indicated in Table 4, the bending moment of the straight segment decreases with an increase in $L_{S}$. For the circular components, the circular ring contracts toward the center under external pressure. In the case of an arch-shaped section, the arc segment contracts toward the center due to external pressure. However, due to the constraints of the straight segment (the two constraints mentioned above) and the fact that the straight segment is not subjected to external pressure, positions closer to the axis of symmetry of the straight segment exhibit smaller deformations. Simultaneously, under the action of the axial force and bending moment, the straight segment deforms along the directions of the *Y*-axis and *X*-axis, with compression deformation along the *Y*-axis direction dominating (Figure 23). This further leads to the “non-rotational symmetry” of the deformation of the arc segment. The displacement contours for the component’s winding layer and liner are similar to those shown in Figure 22 and Figure 23 and are not reiterated here.

Combining Figure 22 and Figure 23, and the preceding analysis, for both the straight segment and the arc segment, their displacements are composed of the compressive displacement $u_{C}^{S}$ (with $u_{C}^{C}$ for the arc segment) induced by axial force (surface pressure for the arc segment) and the bending displacement $u_{M}^{S}$ (with $u_{M}^{C}$ for the arc segment) caused by bending moment. Due to symmetry, it is evident that only radial displacement $u_{C}^{C}(\theta =\pi /2)$ exists on the axis of symmetry for the arc segment, meaning $u_{M}^{C}(\theta =\pi /2)=0$. At this point, $u_{C}^{C}(\theta =\pi /2)$ can be approximately calculated. Assuming the inner radius of the arc section in the arch-shaped liner is $R_{1}$ and the outer radius is $R_{2}$, with the length of the straight segment being $L_{S}$, as revealed by Figure 22 and Figure 23, when $L_{S}/R_{1}\leq 4$, on the axis of symmetry for the arc section at a position with a radius of R, the displacement $u_{C}^{T}(\theta =\pi /2,r=R)$ can be approximated as the sum of the radial displacement $u_{r}^{E}(r=R)$ generated by a circular ring with inner and outer diameters $R_{1}$ and $R_{2}$, respectively, under the same external pressure and the compressive displacement $u_{y}^{E}(y=L_{S}/2,x=R)$ generated by the straight segment under an equivalent axial force and bending moment, i.e.,
(8)$$u_{C}^{T}(\theta =\pi /2,r=R)\approx u_{r}^{E}(r=R)+u_{C}^{S}(y=L_{S}/2,x=R)$$

This argument can be further elucidated through the following physical model. For a liner resembling the non-circular section, during the prestressed winding process, only certain components or positions are subjected to external pressure. This scenario can be simplified and analyzed based on the depicted physical model (Figure 24), where the dashed portion represents the pre-deformation shape, and the solid line represents the post-deformation shape. In this simplification, the region under external pressure is modeled as a plane beam structure, considered as a deformable body, while the non-pressurized part is simplified as a rod, viewed as a rigid body. Consequently, the entire liner is represented as a simply supported beam under surface loads (Figure 24). At this juncture, the displacement of the lowest point of the plane stress beam can be regarded as the combination of compressive deformation $u_{1}^{C}(P)$ and bending deformation $u_{1}^{M}(P)$ induced by the surface load. It is worth noting that when the distance between the two straight segments is small, this deformation can be neglected, that is, we can use the following equation:(9)$$u_{1}^{R}=u_{1}^{C}(P)+u_{1}^{M}(P)$$

Considering the plane beam as a rigid body and treating the non-pressurized part as a pressure-bearing rod, the continuity assumption implies that the displacement of the lowest point on the plane stress beam is consistent with the displacement of the highest point on the pressure-bearing rod (Figure 24), that is, we can imply the following:(10)$$\begin{array}{l} u_{2}^{S}=u_{2}^{C}(y=L_{S}/2)\\ u_{2}^{R}=u_{2}^{S} \end{array}$$

Therefore, the displacement at the lowest point of the plane beam (Figure 24) is
(11)$$u^{R}=u_{1}^{R}+u_{2}^{R}$$

Equations (9)–(11) can be solved using principles from elasticity. As the length of the straight segment has a relatively minor impact on the stress distribution of the arch-shaped component, for non-circular winding, it is feasible to simplify the non-circular section into a combination of several special simple structures. Preliminary analysis can then be conducted using principles from elasticity to draw initial conclusions and identify key factors influencing its mechanical behavior. Based on this foundation, finite element methods are utilized to validate and refine the results from the preliminary analysis, thus allowing us to conduct a more detailed analysis of the key factors.

## 4. Conclusions

In this paper, grounded in the theory of plane beams, we conducted an analysis of an arch-shaped component subjected solely to external pressure on the arc segment. Based on this foundation, a 3D FE model for an arch-shaped prestressed wound component with a metal liner was established. Experimental designs were implemented for comparison, validating the accuracy of the 3D FE model. By comparing the simulation results between the 3D and 2D models, it was concluded that the end effect of the prestressed wound component with an arch-shaped section significantly influences the component’s deformation at the position near the end, but rarely influences those far from the end. Additionally, a comparative analysis was conducted between the prestressed wound components with the arch-shaped sections and those with circular sections to highlight their similarities and differences. According to the analysis mentioned above, a physical model to analyze the stress distribution of prestressed wound component with arch-shaped section is proposed, providing a simplified equivalent method to analyze and calculate the stress and deformation of prestressed wound component with non-circular section.

The main conclusions are the following:Under the condition where only the arc segment of the arch-shaped component is subjected to external pressure, the straight segment will experience the additional bending moment caused by the non-circular symmetry of the structure. This additional bending moment significantly influences the ultimate deformation of the arch-shaped component.Through experiments on the prestressed winding of arch-shaped sections with metal liners, the accuracy of the 3D FE model for prestressed wound components with a metal liner was validated. The error between the simulation results and experimental results is within 2%.By comparing the 3D model with the 2D model, it was observed that the end effect of the arch-shaped component will affect its deformation during the winding process. After a certain distance from the end, the end effect can be neglected. At this point, a 2D model can replace the 3D model for the stress and deformation analysis of prestressed wound components with an arched-shaped liner.By comparing the stress distributions of the arch-shaped section and the circular section for prestressed wound components, it was observed that although the stress in the arc segment of the arch-shaped section exhibits a “higher in the center, smaller at both ends” phenomenon, the stress distribution is very similar to that of the circular section except in the transition zone between the straight segment and the arc segment, with differences between the two being less than 5%. The stress in the straight segment is mainly composed of compressive stress and bending stress. A comparison of the stress distributions in the winding layer revealed that the stress distribution in the arc segment of the component with an arch-shaped section is similar to the component with a circular section, and the differences are negligible. The radial stress in the straight segment is zero, and the circumferential stress is comparable to that in the arc segment.By comparing the deformations of the arch-shaped section and the circular section of prestressed wound components, it was found that when the ratio of the length of the straight segment of the arch-shaped component to the inner diameter of the arc segment is less than 4, the deformation on the symmetric surface of the arc segment of the arch-shaped component can be approximately considered as the superposition of the deformation of a circular section of the same size under the same load. This finding is of significant reference value for analyzing the stress and deformation of wound components with non-circular sections.

## Figures and Tables

**Figure 1 materials-17-00757-f001:**
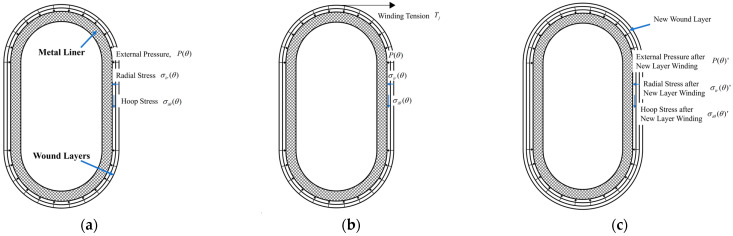
Sketches of the winding procedure of a prestressed wound component with an arch-shaped section: (**a**) before winding; (**b**) during winding; and (**c**) after winding.

**Figure 2 materials-17-00757-f002:**
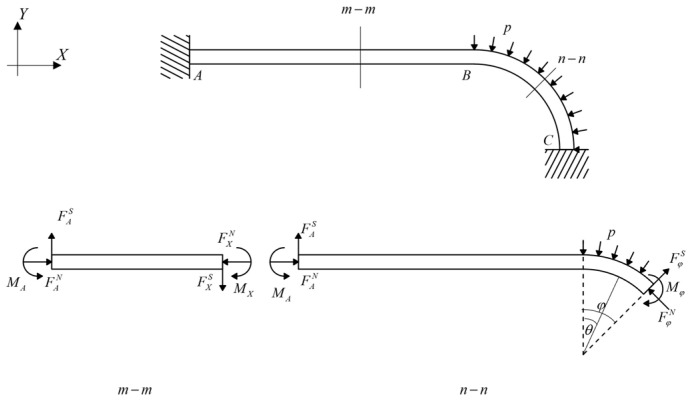
Stress analysis of the prestressed wound liner in an arch-shaped section.

**Figure 3 materials-17-00757-f003:**
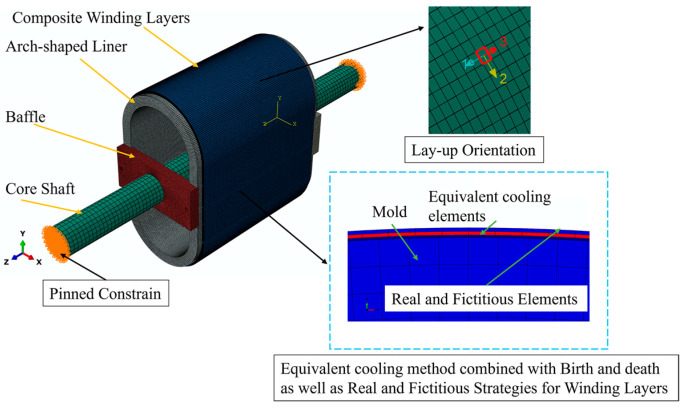
FE model of the prestressed wound component with an arch-shaped metal liner.

**Figure 4 materials-17-00757-f004:**
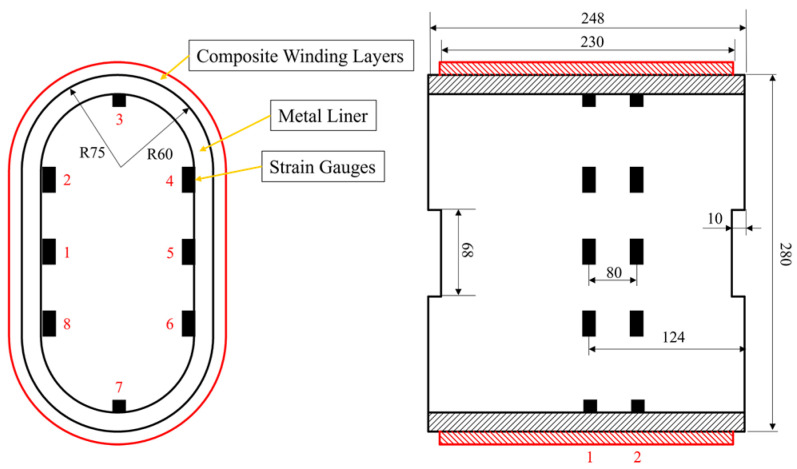
Dimensions of the arch-shaped section component and attachment positions of strain gauges (mm).

**Figure 5 materials-17-00757-f005:**
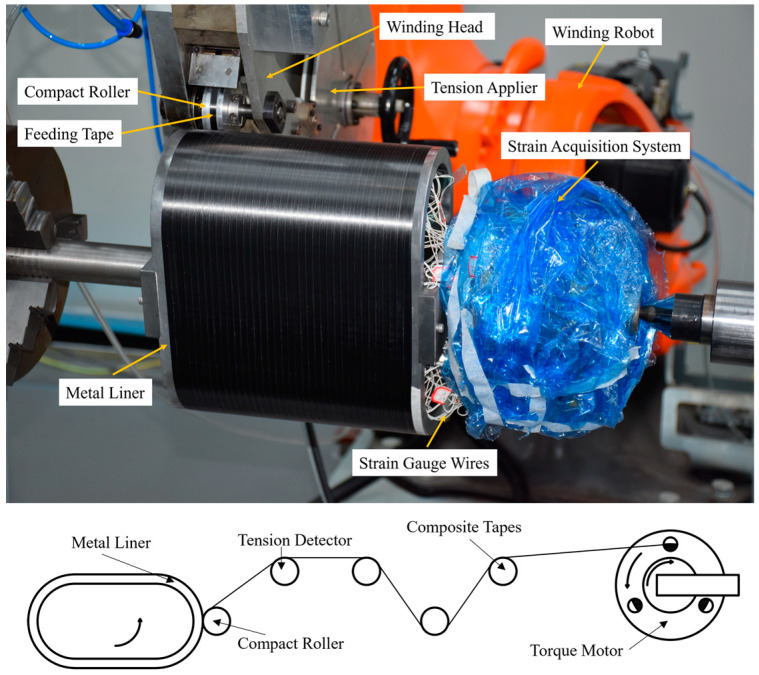
Schematic representation of the experimental setup and process.

**Figure 6 materials-17-00757-f006:**
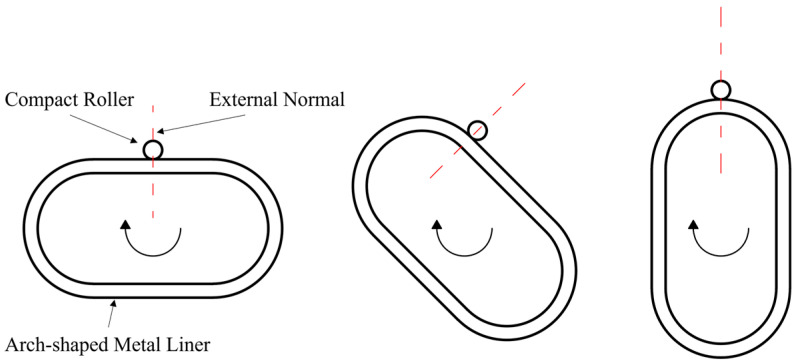
Relationship between the pressure roller and the liner of the arch-shaped section during winding.

**Figure 7 materials-17-00757-f007:**
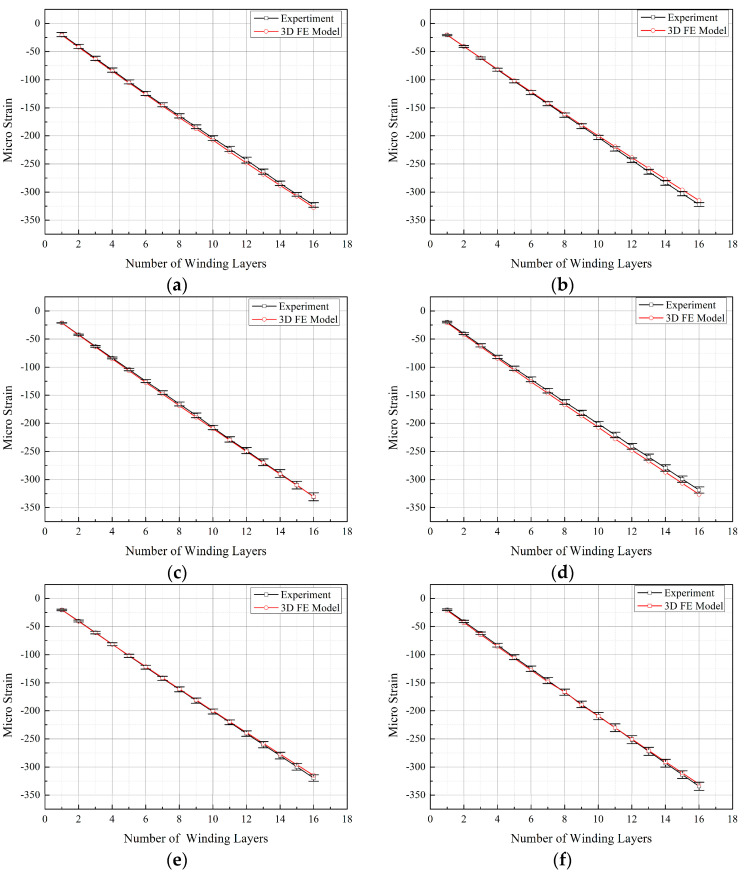
Comparison between experimental results and finite element analysis results. (**a**) Position 1-1. (**b**) Position 1-2. (**c**) Position 1-3. (**d**) Position 2-1. (**e**) Position 2-2. (**f**) Position 2-3.

**Figure 8 materials-17-00757-f008:**
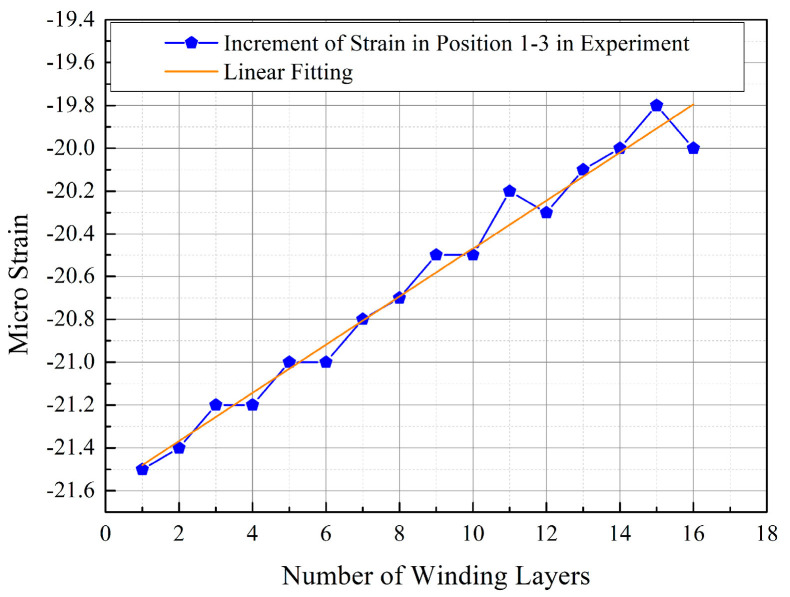
Strain variation at position 1-3 after each winding layer.

**Figure 9 materials-17-00757-f009:**
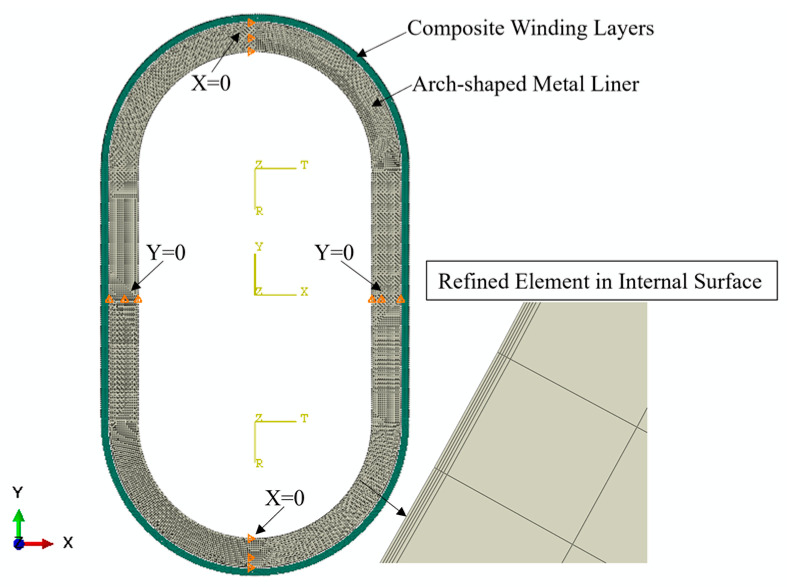
The 2D FE model of the prestressed wound arch-shaped component.

**Figure 10 materials-17-00757-f010:**
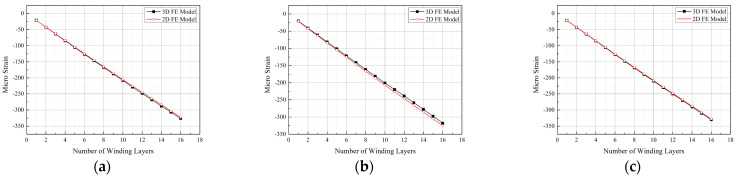
Comparison between the 3D FE model and 2D FE model. (**a**) Position 1-1. (**b**) Position 1-2. (**c**) Position 1-3.

**Figure 11 materials-17-00757-f011:**
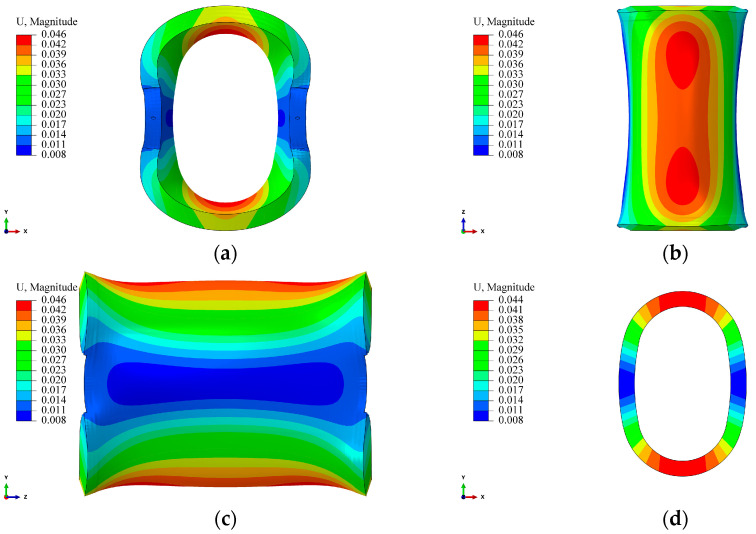
Contour of the arch-shaped liner with prestressed winding finite element deformation. (**a**) 3D FE model—XY plane. (**b**) 3D FE model—XZ plane. (**c**) 3D FE model—YZ plane. (**d**) 2D FE model—XY plane.

**Figure 12 materials-17-00757-f012:**
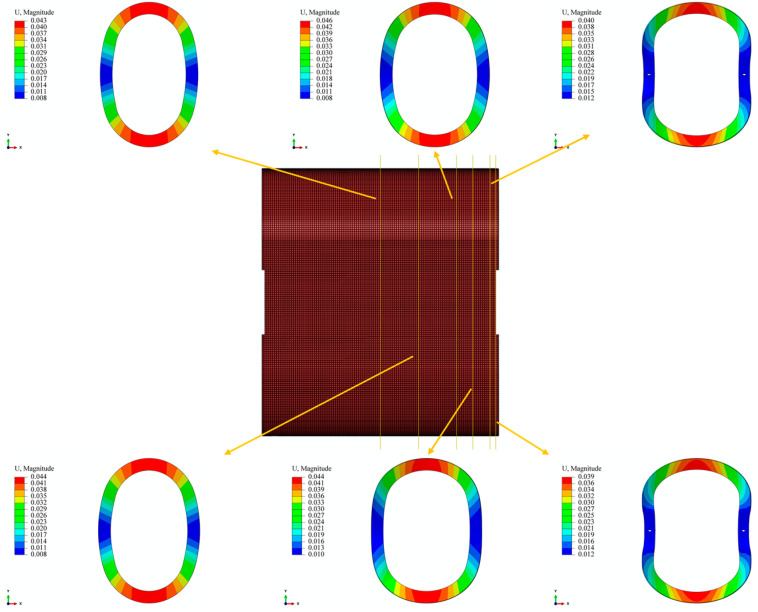
Deformation contour along the axial section of the arch-shaped liner.

**Figure 13 materials-17-00757-f013:**
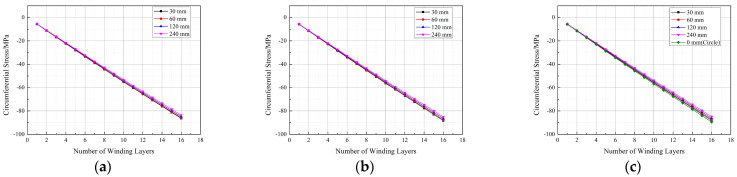
Circumferential stress at measurement points during the winding process of the arch-shaped liner with different lengths of straight segments. (**a**) Position 1-1. (**b**) Position 1-2. (**c**) Position 1-3.

**Figure 14 materials-17-00757-f014:**
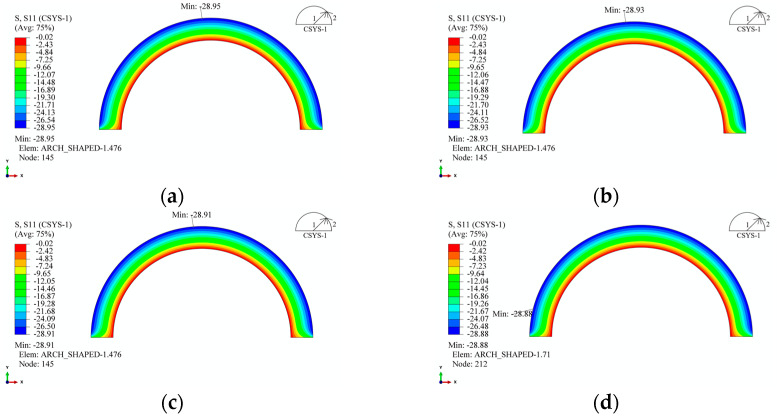
Radial stress distribution of the arch-shaped liner after winding. (**a**) 30 mm. (**b**) 60 mm. (**c**) 120 mm. (**d**) 240 mm.

**Figure 15 materials-17-00757-f015:**
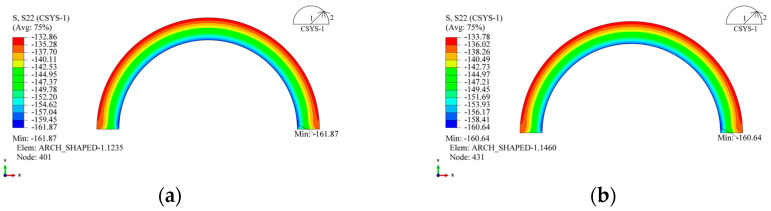
Circumferential stress distribution of the arch-shaped liner after winding. (**a**) 30 mm. (**b**) 60 mm. (**c**) 120 mm. (**d**) 240 mm.

**Figure 16 materials-17-00757-f016:**
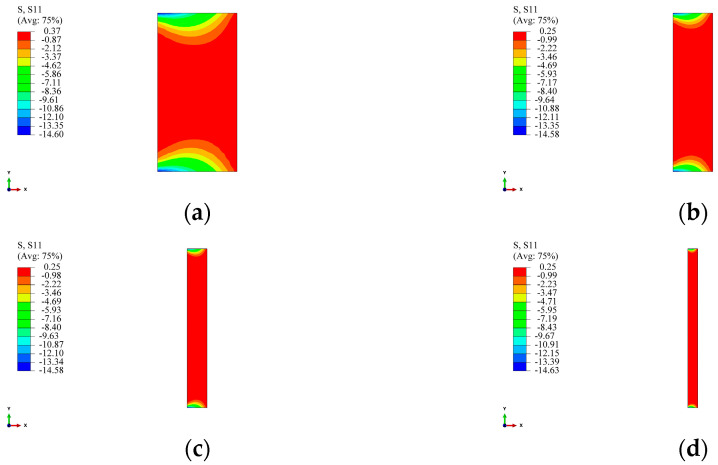
*X*-axis stress in the straight segment of the arch-shaped liner. (**a**) 30 mm. (**b**) 60 mm. (**c**) 120 mm. (**d**) 240 mm.

**Figure 17 materials-17-00757-f017:**
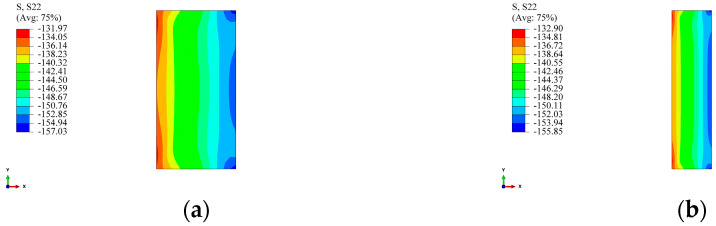
*Y*-axis stress in the straight segment of the arch-shaped liner. (**a**) 30 mm. (**b**) 60 mm. (**c**) 120 mm. (**d**) 240 mm.

**Figure 18 materials-17-00757-f018:**
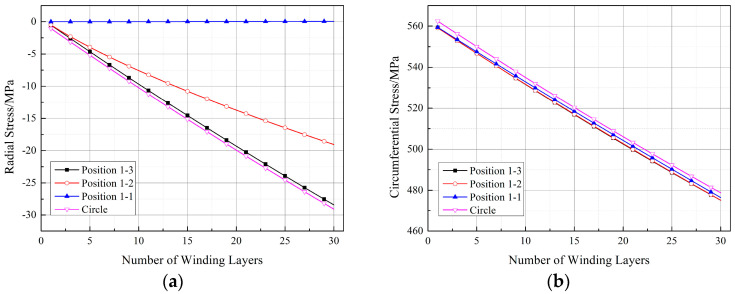
Stresses at the positions corresponding to the measurement points of the first winding layer (straight segment length: 30 mm): (**a**) radial stress and (**b**) circumferential stress.

**Figure 19 materials-17-00757-f019:**
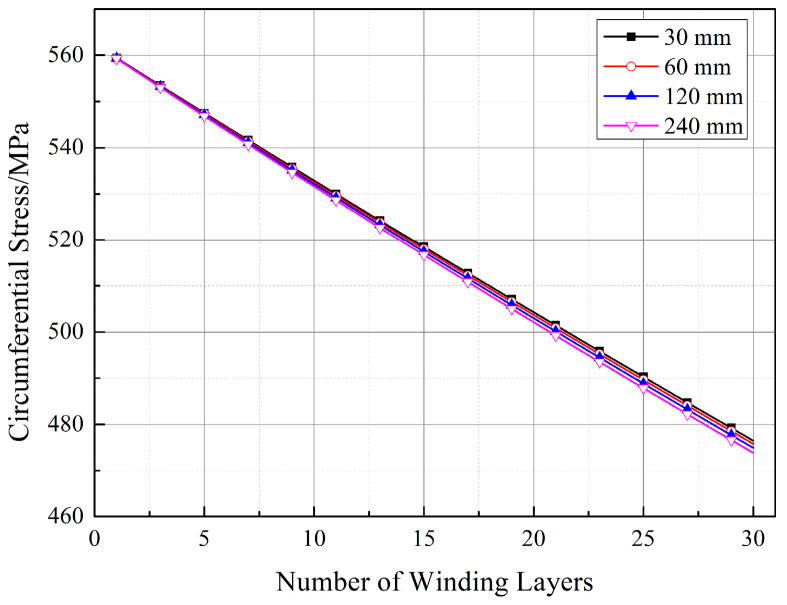
Stresses in the winding layer for different lengths of straight segments (position 1-1).

**Figure 20 materials-17-00757-f020:**
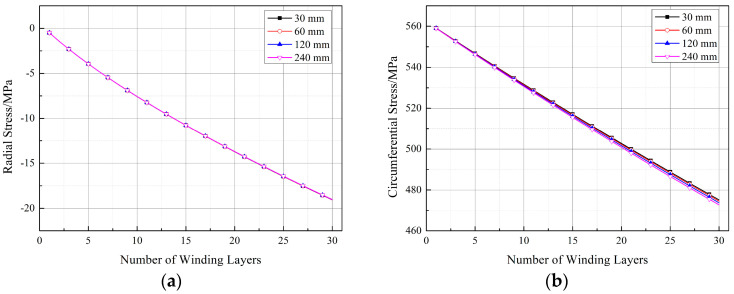
Stresses in the winding layer for different lengths of straight segments (position 1-2): (**a**) radial stress and (**b**) circumferential stress.

**Figure 21 materials-17-00757-f021:**
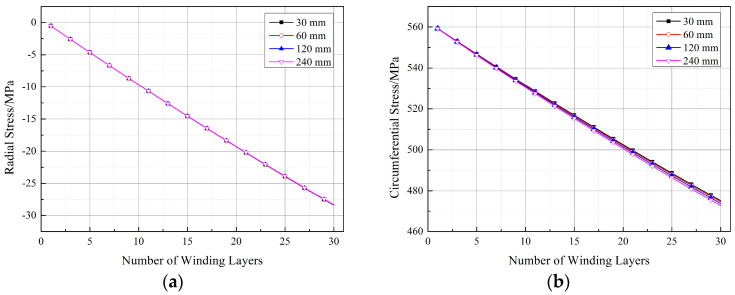
Stresses in the winding layer for different lengths of straight segments (position 1-3): (**a**) radial stress and (**b**) circumferential stress.

**Figure 22 materials-17-00757-f022:**
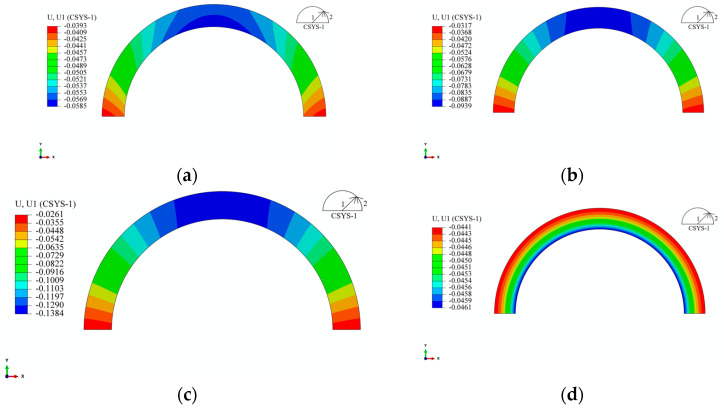
Radial displacement of the arc segment of the arch-shaped liner after winding. (**a**) 30 mm. (**b**) 120 mm. (**c**) 240 mm. (**d**) Circle.

**Figure 23 materials-17-00757-f023:**
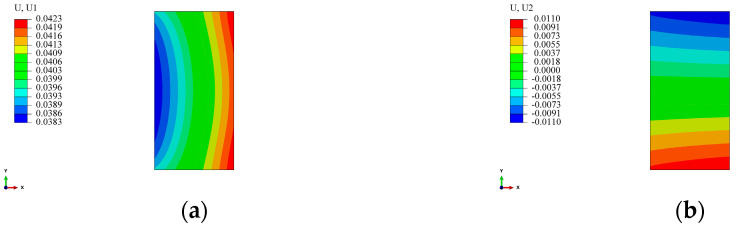
Displacement of the straight segment of the arch-shaped liner after winding. (**a**) 30 mm—Direction X. (**b**) 30 mm—Direction Y. (**c**) 120 mm—Direction X. (**d**) 120 mm—Direction Y. (**e**) 240 mm—Direction X. (**f**) 240 mm—Direction Y.

**Figure 24 materials-17-00757-f024:**
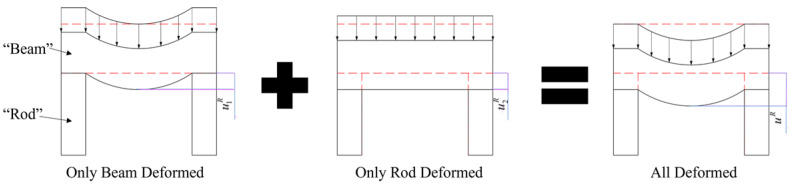
Physical model of the deformation of non-circular wound components (dashed potion for pre-deformation shape, and solid line for post-deformation shape).

**Table 1 materials-17-00757-t001:** Dimensional parameters of the FE model for the prestressed wound component with an arch-shaped section.

Parameter	Value
Inner radius of the arc segment in the liner of the arch-shaped section, $R_{1}$/mm	60
Outer radius of the arc segment in the liner of the arch-shaped section, $R_{2}$/mm	75
Length of the straight segment in the liner of the arch-shaped section, $L_{S}$/mm	130
Total number of winding layers, $L_{T}$	16
Thickness of a single winding layer, t/mm	0.14
Width of a single winding layer, h/mm	6.35
Winding tension, T/N	400

**Table 2 materials-17-00757-t002:** Material properties of Q235.

Material	Modulus/GPa	Poisson’s Ratio
Q235	210	0.34

**Table 3 materials-17-00757-t003:** Material properties of the composite AS4D/PEEK.

Parameter	Value
Fiber longitudinal elastic modulus, $E_{1}$/GPa	133
Transverse elastic modulus, $E_{2}$/GPa	9.7
Elastic modulus in the thickness direction, $E_{3}$/GPa	9.7
Poisson’s ratio, $\mu_{12}$	0.29
Poisson’s ratio, $\mu_{13}$	0.29
Poisson’s ratio, $\mu_{23}$	0.3
Shear modulus in the 1–2 plane, $G_{12}$/GPa	5.5
Shear modulus in the 1–3 plane, $G_{13}$/GPa	5.5
Shear modulus in the 2–3 plane, $G_{23}$/GPa	3.35

**Table 4 materials-17-00757-t004:** Bending moment ($M_{A}$) at the midpoint of the straight segment with a different length.

$L_{s}$/mm	Additional Bending Moment/(N mm)
30	339.9
60	311.4
120	256.9
240	189.2

## Data Availability

Data are contained within the article.

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
