# Peer review of "Stress and Deformation Analysis of Prestressed Wound Composite Components with an Arch-Shaped Metal Liner"

_materials, 2024, doi:10.3390/ma17030757_

Round 1
Reviewer 1 Report
Comments and Suggestions for Authors
This article discusses the analysis conducted on prestressed winding components with metal liners featuring arch-shaped sections. It involves some related analytical formulations, experimental tests and numerical analysis. The reviewer finds that the topic can have some interest for the research community, though most of the analytical formulations are obvious from solid mechanics literature.
Though the scope of the research is relatively narrow, the article is too large due to some unnecessarily extended descriptions. The article rather sounds a thesis work, or it can be a chapter extracted from a thesis, which is not properly suited to a journal article. The sentence "The main conclusions of this chapter are as follows:" in the conclusion section indicates this fact.
In the reviewer's opinion, the last paragraph of the introduction must be moved to the the Materials and Methods section. Furthermore, the materials and methods section is presented on about 7 pages to present the analytical approach, FE modelling technique and the experimental setup in Sections 2.1, 2.2 and 2.3 respectively. These sections must be reduced to less than half of the current size. Section 2.1 in particular contains many basic contents and formulas, even which are not directly used in the discussion of the results or no analytical calculations are done and reported using these equations. The reviewer recommends also to to modify the headings of these sections as
2.1. Analysis of Stress Distribution
2.2. FE modelling
2.3. Experimental setup
Some specific comments
1. The way the author affiliations are given must be corrected. As I see, all authors have the same affiliation, only different e-mails.
2. Wordings like "comprehensive understanding (in abstract)", "comprehensive finite element analysis (Line 88)", "relatively comprehensive (Line 123" must be revised. What does comprehensive analysis or understanding mean?
3. Too few and very generic keywords are used.
4. Line 42: "... reaching up to 50% of the material strength." needs reference.
5. Line 86: "... finite element analysis stands out as a critical approach ..." What do you mean?
6. Line 88: "... combines virtual and real elements ..." Please explain.
7. Table 4: Since all values are average, what is the purpose of have the superscript A? Figure 4 contains duplicate information of Table 4. Use only either table or figures.
8. Section 3.2.2: Please justify the need to simulate 2D and 3D. Which one is most proper for validation with the experiment?
9. Table 5: Is it really "Material property" which is given in the table?
Comments on the Quality of English Language
The article must be significantly reduced to grasp the research novelty.
Author Response
Dear Reviewer,
Thank you very much for taking the time to review this manuscript. We have revised our manuscript according to your comment. The point-by-pont response has been put on the Word file uoloaded. Please see the attachment.

Reviewer 2 Report
Comments and Suggestions for Authors
This paper introduces a physical model to analyze stress and deformation in prestressed filament fiber-wound components with arch-shaped sections, exploring the additional bending moment effect. A 3D finite element model validates the study's findings.
The paper can be considered for publication in Materials provided the authors address the minor points raised below:
Figure 1 and all subsequent figures should have enhanced captions, offering detailed explanations for a more comprehensive understanding. Captions must be self-explanatory to facilitate reader comprehension.
Is the strain presented in percentage in the charts shown? If so, this detail should be explicitly mentioned in the text, as well as in all pertinent tables and figures displaying macrostrain values.
The quality of the figures is low and should be improved.
Are the S11 and Sxx the same? 11 and 22 directions should be shown in the figures with FE results.
Author Response

(The authors gave the same response as above.)

Reviewer 3 Report
Comments and Suggestions for Authors
Paper is well written, and I am going to accept the paper after considering the following comments:
1) In Figure 3, please explain more about the details especially the applied loads and other BCs.
2) Please increase the quality of every figure.
3) Please mention explicitly what the novelty of your work is.
Author Response

(The authors gave the same response as above.)

Round 2
Reviewer 2 Report
Comments and Suggestions for Authors
-